# Comparative Analysis of the Floral Fragrance Compounds of *Panax notoginseng* Flowers under the *Panax notoginseng*-*pinus* Agroforestry System Using SPME-GC-MS

**DOI:** 10.3390/molecules27113565

**Published:** 2022-06-01

**Authors:** Siyu Chen, Rui Rui, Shu Wang, Xiahong He

**Affiliations:** 1Key Laboratory for Forest Resources Conservation and Utilization in the Southwest Mountains of China, Ministry of Education, Southwest Forestry University, Kunming 650224, China; 15846027621@163.com (S.C.); ruiruiswfu@163.com (R.R.); 2Department of Horticulture, College of Landscape Architecture and Horticulture, Southwest Forestry University, Kunming 650224, China; 3Southwest Landscape Architecture Engineering Research Center of National Forestry and Grassland Administration, Kunming 650224, China

**Keywords:** *Panax notoginseng*, floral scent compounds, headspace solid-phase microextraction, gas chromatography–mass spectrometry, cluster analysis

## Abstract

*Panax notoginseng* is a medicinal plant in China, the flowers of which have high medicinal value. To study the differences in the floral fragrance compounds of *P. notoginseng* flowers (bionic wild cultivation) from the forests of Yunnan Province, the floral fragrance compounds from four varieties of *P. notoginseng* flowers (four-forked seven leaves, three-forked seven leaves, four-forked five–seven leaves, and three-forked five–six leaves) were compared and analyzed via headspace solid phase microextraction combined with gas chromatography–mass spectrometry methods. A total of 53 floral fragrance compounds from the *P. notoginseng* flowers were divided into eight categories, mainly consisting of terpenes, alkynes, aromatic hydrocarbons, and alcohols. Moreover, high contents of 3-carene, germacrene D, (−)-α-gurjunene, valencene, (+)-*γ*-gurjunene, menogene, and aromandendrene were identified from the flowers of different *P. notoginseng* varieties. Interestingly, floral fragrance compounds such as 3-carene, valencene, aromandendrene, menogene, and (+)-*γ*-gurjunene were first reported in the flowers of *P. notoginseng*. Cluster analysis showed that *P. notoginseng* with four-forked and three-forked leaves clustered into two subgroups, respectively. In addition, principal component analysis showed that (+)-*γ*-gurjunene, (+)-calarene, copaene, 1,8,12-bisabolatriene, *γ*-elemene, (–)-aristolene, caryophyllene, 3-carenes, and 2,6-dimethyl-1,3,6-heptatriene can be used to distinguish the floral fragrance components of four *P. notoginseng* flower species. This study provides a theoretical basis for elucidating the floral fragrance compounds emitted from the flowers of different *P. notoginseng* varieties in an agroforestry system.

## 1. Introduction

*Panax notoginseng*, also known as Tianqi and Sanqi, is a perennial herb that belongs to the Araliaceae family [1]. *P. notoginseng* is a precious medicinal plant in China, and it mainly grows in some areas of the Yunnan and Guangxi provinces. The flowers of *P. notoginseng* are valuable medicinal material, and they function in clearing away heat, in detoxification, in calming the liver, in improving eyesight and in the production of saliva, and in the quenching of thirst [2]. These functions are a result of the variety of active components that are present in *P. notoginseng* flowers, such as olefin compounds [3,4], ginsenoside (Rb1, RC, RB3, Rb2, and Rd) [5,6], and polyphenols. Moreover, the polyphenol content in *P. notoginseng* flowers is significantly higher than the content in the *P. notoginseng* root [7], where polyphenols have strong antioxidant activity [8], function in the protection of the cardiovascular and cerebrovascular systems, have anti-tumor properties, and assist in the prevention of neurological diseases [9]. In addition, *P. notoginseng* can be used as an additive in food [10], as a material for the development of new drugs [11], for tea beverages [12], and even in shampoo [12]. Therefore, a study on the floral fragrance compounds of *P. notoginseng* flowers can provide a theoretical foundation for the processing of edible products, thus aiding in the development and utilization of medicines.

There were significant differences in the floral components of different medicinal plants with different species. Studies have shown that there are 30, 31, 31, and 20 kinds of aroma components in four different varieties of *Dendrobium* orchid (*Dendrobium aphyllum*, *D. nobile*, *D. candidum*, and *D. loddigesii*), respectively, and although the common floral fragrance components mainly consist of trans-α-ocimene and *β*-myrcene [13,14,15], the contents of the volatile substances are different. Similar results have shown that the types and contents of the floral fragrance components in different varieties of lotus (Honghu red lotus, Hongwanwan, Hongpeony, Baiwanwan, and yellow peony) were also different, i.e., 33, 34, 33, and 32, with the highest contents being alkanes, olefins, and alcohols [16]. In addition, the researchers studied the floral fragrance components of six varieties of peony (*Peony’Zhaofen*, Luoyanghong, Fengdanbai, Zhongwu, High noon, Renown, and Gaoyuanshenghuo’) and found significant differences in the total amount of floral fragrance released by the six varieties, which were 48, 45, 44, 44, 28, and 40 compounds, respectively [17]. Furthermore, 19, 18, 11, and 20 floral fragrance compounds were detected in four varieties of wax plums with different fragrance types in Yunnan, the main components of which were *α*-ocimene, benzyl alcohol, benzyl acetate, and eugenol [18]. The eight cultivars also differed in the aromas (white, pink, and red flowers) of plum blossoms of different colors, with 18–23 aroma components [19]. As mentioned above, the flowers of the different plant varieties, floral fragrance types, and flower colors are closely related to the floral components of the plants.

Light, temperature, rhythm, and different environments, as well as the plant variety, all play roles in the fragrance of plant flowers. Zou Jingjing [20] and others found that the aroma substances and the related chemical content in *Osmanthus fragrans* changed significantly under light and dark treatments, and continuous light and dark conditions were able to reduce the release of aroma-active substances. Therefore, light is an important factor that affects the floral fragrance of plants. At the same time, differences in temperature also affect the floral aroma components and types. Studies have shown that with an increase in temperature (10, 20, and 30 °C), the types of aroma components in *Oncidium* gradually increase (24, 33, and 43) [21]. As far as the floral components are concerned, the content of sweet-scented *Osmanthus* components (ketones, monoterpenes, and oxidized derivatives) increased at lower temperatures (4 °C). On the contrary, the content of esters increased with an increase in temperature (45 °C) [20]. In addition, under different rhythm conditions, the number and content of floral components showed a trend of first increasing and then decreasing; this was the case with lily [22] and *Osmanthus* [23]. Similarly, there were significant differences in the floral aroma compounds of plants from different geographical regions. For example, the same variety of wax plum blossom has different floral aroma compounds in Yunnan [18], Zhejiang [24], Jiangxi [25], Sichuan [26], and other regions. As mentioned above, the light, temperature, rhythm, and different environmental conditions have important effects on the floral fragrance. Compared with the environment in which *P.*
*notoginseng* is grown within traditional agriculture, the growth environment of bionic wild-cultivated *P.*
*notoginseng* under pine trees is significantly different, and therefore, this may affect the floral fragrance compounds of *P.*
*notoginseng* flowers. However, there have been no reports of changes in the floral composition in flowers of different *P**. notoginseng* varieties grown under pine trees.

After 2015, the agroforestry model of Pine-Sanqi was planted in large quantities in Yunnan (Pu’er, Lincang, Kunming, Qujing, etc.), with an area of approximately 1500 ha. The flowers of *P. notoginseng* in a conventional agricultural system have a slightly bitter odor, but the flowers of *P. notoginseng* in the agroforestry system have a different odor, which may be due to environmental factors resulting in the production of some special compounds. Therefore, this study aims to analyze and identify the floral fragrance compounds of the native flowers of four species of *P. notoginseng* in the agroforestry system in Yunnan via solid phase microextraction (SPME)/gas chromatography–mass spectrometry (GC-MS) technology, to evaluate and screen valuable *P. notoginseng* aromatic germplasm resources, and to provide a theoretical and practical basis for the cultivation of *P. notoginseng* varieties and the development and utilization of future products.

## 2. Results

### 2.1. Comparative Analysis of Categories for Floral Fragrance Compounds in the Flowers of Different Varieties of P. notoginseng

The floral scent compounds emitted by the flowers of four *P. notoginseng* plant varieties are shown in Table 1. Among the four varieties of *P. notoginseng* in Yunnan, S1, S2, S3, and S4 (four-forked seven leaves, three-forked seven leaves, four-forked five–seven leaves, and three-forked five–six leaves, respectively), a total of eight categories of floral fragrance compounds were detected, including terpenes, alcohols, esters, aromatic hydrocarbons, alkynes, alkanes, aldehydes, and others. Moreover, high-content floral fragrance compounds included terpenes (71.65–84.16%), alcohols (0.31–1.15%), aromatic hydrocarbons (8.87–10.61%), alkynes (2.82–15.85%), and others (0.11–1.17%). However, esters were only detected in S3 (0.34%) and alkanes were only detected in S1 and S3, with contents of 0.88% and 1.00%, respectively. Additionally, aldehyde was the only compound not detected in S2.

### 2.2. Analysis of Floral Fragrance Compounds in the Flowers of Different Varieties of P. notoginseng

A total of 53 floral fragrance compounds were identified in four varieties of *P. notoginseng* flowers, of which 35, 36, 39, and 35 components were detected in the varieties S1, S2, S3, and S4, respectively (Table 2), with relatively high contents of 3-carene, menogene, (−)-α-gurjunene, germacrene D, (+)-*γ*-gurhunene, and 2,4-diethyl-2,3-octadien-5-in. For each category, there was a high content of included terpenes (germacrene D at 11.47–27.25%, 3-carene at 15.63–25.69%, and (−)-α-gurjunene at 12.22–12.67%), aromatic hydrocarbons (+)-*γ*-gurhunene at 5.33–10.56%, and alkynes 2,4-diethyl-2,3-octadiene-5-in at 2.94–4.04%.

### 2.3. Analysis of Floral Fragrance Compounds (>1%) in the Flowers of Different Varieties of P. notoginseng

Figure 1 presents a total of 24 floral fragrance compounds (>1%) in *P. notoginseng* flowers, including 3-carene, germacrene D, valencene, (−)-*α*-gurjunene, (+)-*γ*-gurjunene, aromandendrene, menogene, 2,4-diethyl-2,3-octadiene-5-in, trans-alloocimene, 2,6-dimethyl-1,3,6-heptatriene, (+)-calarene, copaene, *β*-cadinene, terpinolene, caryophyllene, (−)-*α*-pinene, *γ*-elemene, (−)-aristolene, *β*-copaene, 2-(3-methyl-but-ynyl)-cyclohexene-1-carboxaldehyde, cis-muurola-3,5-diene, 1,8,12-bisabolatriene, 5,6-decadien-3-yne,5,7-diethyl, and (−)-*α*-muurolene. Moreover, there were high levels of 3-carene (15.63–25.69%), germacrene D (11.47–27.25%), valencene (1.17–14.44%), and (−)-*α*-gurjunene (12.22–12.67%) in the flowers of different *P. notoginseng* varieties. In addition, some *P. notoginseng* varieties included special floral fragrance compounds. For example, cis-muurola-3,5-diene and (−)-*α*-muurolene were only detected in the flowers of S4, and 5,6-decadien-3-yne,5,7-diethyl was only detected in the flowers of S2. Floral fragrance compounds such as *β*-copaene, 2-(3-methyl-but-ynyl)-cyclohexene-1-carboxaldehyde, *γ*-Elemene, and 1,8,12-bisabolatriene were absent in S2 and S4.

### 2.4. Hierarchical Cluster Analysis of Floral Fragrance Compounds in the Flowers of P. notoginseng

To explore the relationship between the floral fragrance compounds and different *P. notoginseng* varieties within an agroforestry system, hierarchical cluster analysis was performed on the main floral fragrance compounds (>1%) in the flowers of *P. notoginseng* (S1, S2, S3, and S4). As shown in Figure 2, *β*-cadinene, caryophyllene, (−)-aristolene, and copaene in S1 and S2 were similar and clustered into one subgroup. In addition, a high content of valencene, cis-muurola-3,5-diene, (+)- *γ*-gurjunene in S4, and (+)-calarene in S3 was detected; hence, S3 and S4 clustered into two subgroups.

### 2.5. Principal Component Analysis of Floral Fragrance Components in the Flowers of P. notoginseng

As shown in Figure 3, principal component analysis was conducted on the floral fragrance components of the flowers of *P. notoginseng* within an agroforestry system. The variances of PC1, PC2, and PC3 were 42.319%, 34.263%, and 23.418%, respectively. The highest loading values in each PC were used as the main factor, with five compounds having the highest loading values in PC1, namely (+)-*γ*-gurjunene, (+)-calarene, copaene, 1,8,12-bisabolatriene, and *γ*-elemene. In addition, (−)-aristolene, caryophyllene, 3-carenes, and 2,6-dimethyl-1,3,6-heptatriene were the four substances that contributed the most to PC2 and PC3. As a result, the three compounds with the highest loading values in each of the three PCs were selected as the main factors, which showed that (+)-*γ*-gurjunene, (+)-calarene, copaene, 1,8,12-bisabolatriene, and *γ*-elemene indole contributed the most to the floral fragrance compounds. Therefore, the variables were divided into (+)-*γ*-gurjunene, (+)-calarene, copaene, 1,8,12-bisabolatriene, *γ*-elemene, (−)-aristolene, caryophyllene, (−)-aristolene, caryophyllene, 3-carenes, and 2,6-dimethyl-1,3,6-heptatriene, as they were also representative floral fragrance compounds of *P. notoginseng* within the different varieties in the agroforestry system.

## 3. Discussion

Flower fragrance is a secondary metabolite that contains particular chemical information; it consists of various low molecular weight compounds and mixtures with low boiling points [27]. Flower fragrance is an important component of plant volatile compounds [28] that are released from plants [29]. Presently, most previous studies have focused on the identification of volatile oil components of the *P. notoginseng* flower, but the identification of the floral fragrance compounds of the *P. notoginseng* flower have not been reported. In this study, the floral fragrance compounds of four species of *P. notoginseng* flowers were identified, and a total of 53 compounds were detected in the floral fragrance compounds of *P. notoginseng* flowers, which were similar to the volatile oil components (55) of *P. notoginseng* flowers in Wenshan, Yunnan [30]. However, the components of their volatile compounds exhibited differences. In addition, inconsistent findings have shown that volatiles vary significantly in terms of quantity and composition; for example, 24 and 37 volatile oil components in *P. notoginseng* flowers and 34 volatile oil components in *P. notoginseng* plants [31,32,33,34]. Therefore, it is evident that there is an inconsistency between the floral compounds of the *P. notoginseng* flower and the compounds in the volatile oils of the *P. notoginseng* flower.

The major components of the floral fragrance compounds of *P. notoginseng* in the forest include terpenes (30), alcohols (5), and hydrocarbons (11). This result is similar to that of *P. notoginseng* in Wenshan, Yunnan, where the volatile oil contains mainly terpenes (terpenes and their oxygenated derivatives) [30]. However, this is different from the results of the previous study, where the main components of volatile oils in *P. notoginseng* flowers in Guangxi and Yunnan Wenshan are terpenes (3, 6), hydrocarbons (12, 6) [31], and esters. In addition, many studies have shown that the main components isolated from the volatile oils of *P. notoginseng* are monoterpenes, sesquiterpenes, and their derivatives [32,33,34]. As mentioned above, it is suggested that terpenes are one of the main floral fragrance compounds and volatile oils in the *P. notoginseng* flower. Interestingly, a high level of terpenes (3-carene, valencene, aromandendrene, and menogene) in the floral fragrance compounds was not detected in the volatile oils of *P. notoginseng* flowers from the Yunnan and Guangxi provinces [30,31]. The terpenes (−)-*α*-gurjunene and germacrene D were identified in the volatile oils of *P. notoginseng* flowers in Wenshan, Yunnan [32]. In addition, the contents of some terpenoids (*α*-guaiene, alloaromadendrene, and *γ*-elemene) were lower in the floral compounds of *P.*
*notoginseng* flowers and higher in the volatile oils of *P.*
*notoginseng* [34]. These results were in accordance with the previous study, where the floral fragrance compounds of *Michelia maudiae* were different from its volatile oil compounds [35]. Although the floral fragrance compounds and the volatile oil components of the *P.*
*notoginseng* flowers from the forest understory had some common volatiles, new floral fragrance compounds, including 3-carene, valencene, aromandendrene, menogene, and (+)-γ-gurjunene, were observed in our study. This may be due to the bionic wild cultivation of *P.*
*notoginseng* under pine trees without the application of fertilizers and pesticides, which are different from *P.*
*notoginseng* plantations in traditional agriculture. Therefore, we speculate that the pattern of cultivation, and the fertilizers and pesticides applied, had an effect on the floral fragrance compounds of the flowers. Furthermore, the light and temperature of the agroforestry system have changed with land-use conversion processes, which has also affected the floral fragrance compounds of Sanqi flowers. Previous results have shown that both light and temperature have effects on the fragrance component of *Osmanthus fragrans* [20]. In addition, the floral fragrance compounds were affected by the geographical regions. For example, different floral fragrance compounds were observed in *Chimonanthus precox* in Yunnan, Zhejiang, Jiangxi, and Sichuan [18,24,25,26]. In summary, light, temperature, and environment are the main factors that result in variations of floral fragrance components.

Previous studies have shown that there are large differences in floral fragrance components between different plant varieties, especially with regard to flower shapes, colors, and aroma types. A high content of floral fragrance compounds, including 3-carene (15.63–25.69%), germacrene D (11.47–27.25%), valencene (1.17–14.44%), and (−)-*α*-gurjunene (12.22–12.67%) were found in different varieties of *P. notoginseng* flowers. However, some floral fragrance compounds, such as cis-muurola-3,5-diene (S4), (−)-*α*-muurolene (S4), *β*-copaene (S1, S3), and 5,6-decadien-3-yne,5,7-diethyl (S2), were detected only in certain varieties. The main differences between different species of *P. notoginseng* are that, although the flower morphology is the same, the numbers of forks and leaves are different, and these differences may lead to differences in the floral fragrance compounds of *P. notoginseng* flowers. Based on transcriptome analysis, different varieties of peony [27] and lily [28] differ in terms of the related genes that are involved in floral fragrance, while the synthesis of floral fragrance compounds is mainly influenced by the expression of related genes, the regulation of enzyme activities, and the involvement of biosynthetic pathways [36]. Therefore, we speculate that different genes that are involved in the pathways related to floral fragrance synthesis in different varieties may lead to the differences in floral fragrance composition. Floral fragrance compounds in the flowers of *P. notoginseng* in an agroforestry system contain some active components. For example, a high content of 3-carene, which is a natural monoterpene, has been shown to confer antibacterial activity [37]. Germacrene D and spatulenol regulate defense and pollination, and anti-inflammatory effects, respectively [38]. A low content of caryophyllene and *β*-cadiene in floral fragrance compounds with anti-asthmatic, antitussive, and expectorant effects [30] is consistent with the effect of *P. notoginseng* flowers on the treatment of pharyngitis [39]. In addition, some unique floral fragrance compounds, such as *α*-guaiene and *β*-copaene, are linked to the relief of coughs and the reduction in phlegm [30] and to antioxidant activity, respectively [40]. In our next steps of the study, we will analyze and test the pharmacological activities of the main floral fragrance compounds in *P**. notoginseng* flowers, as this will provide a basis for future applications.

## 4. Materials and Methods

### 4.1. Plant Materials

Plant materials were selected from *P. notoginseng* varieties grown in a forest in Xundian County, Kunming City, Yunnan Province (103°12′45″ E, 25°28′18″ N). In January 2019, 1-year-old *P. notoginseng* seedlings were selected and transplanted into the forest for growth. Weeds were first removed from the soil where *P. notoginseng* was planted. After tillage, *P. notoginseng* was planted with a planting density of 5 × 5 cm. The slope generally varied between 5 and 15°, and the monopoly width was approximately 120 cm. The planting regulations of *P. notoginseng-pinus* in an agroforestry system are shown in Figure 4. *P. notoginseng* with different species were planted as follows: a four–fork seven leaf (four palmately compound leaves and seven leaves), a three-fork seven leaf (three palmately compound leaves and seven leaves), a four–fork five–seven leaf (four palmately compound leaves, one of which consists of five leaves and the other three consisting of seven leaves), and a three–fork five leaf (three palmately compound leaves, one of which has six leaves and the other two have five leaves)*,* which were numbered S1, S2, S3, and S4, respectively. The morphological characteristics of the four varieties are shown in Figure 5.

### 4.2. Sample Collection

At 10:00 am on 10 August 2021, flowers from different *P. notoginseng* varieties (biennial) with healthy growth and no disease were selected for experiment. The experiment was set up in three repetitions, for a total of 12 bottles. The flowers (five flowers/bottle) were sampled and then weighed and placed into a 20 mL SPME bottle (clear glass, flat bottom; 20 mm open seal with PTFE/silicone septa, aluminum crush cap, Thermo Fisher Scientific, Waltham, MA, USA). Subsequently, the samples were sealed with sealing pliers for floral fragrance analysis.

### 4.3. HS-SPME Analysis

The optimization of SPME-GC/MS based on the methods of *Chimonanthus praecox* was performed in our study [18]. A 100 μm PDMS SPME fiber (American Supelco Company) was combined with an SPME automatic sampler device for HS-SPME analysis. After the sample was equilibrated at room temperature for 30 min, the floral fragrance substances of *the P. notoginseng* flower were adsorbed into the extraction fiber. Before sampling the floral fragrance substance, the GC-MS inlet of the SPME fiber was adjusted to 250 °C for 40 min. The fiber was then inserted into the top of a sealed SPME vial using an SPME autosampler (fiber conditioning temperature of 250 °C, 20 min, TriPlus autosampler, Thermo Fisher Scientific), and the sample was adsorbed at room temperature for 40 min.

### 4.4. GC-MS Analysis

When the adsorption was complete, the extraction head was withdrawn, inserted into the GC-MS injection port (Trace GC Ultra/ITQ 900, Thermo Fisher Scientific), and dissociated at 250 °C for 20 min, and then the instrument was initiated to collect the data. The GC conditions were as follows: an HP-5MS capillary column with a length of 30 m was used as the chromatographic column, the carrier gas was high-purity helium (99.999%), the flow rate was 1.0 mL/min, and the sample volume was 1 µL without a shunt. The temperature rise procedure was as follows: the injection port temperature was 250 °C, and the starting temperature of the column was 50 °C; this was held for 4 min. The temperature was raised to 150 °C at 10 °C/min, held for 15 min, and then raised to 250 °C at 2 °C/min, and held for 11 min. The MS conditions were as follows: an EI ion source, an ionization energy of 70 eV, an ion source temperature of 230 °C, and a mass scanning range of 50–550 amu.

### 4.5. Data Analysis

NIST14 (National Institute of Standards and Technology) mass spectrometry database was used for qualitative identification, and only the results with positive and negative matching values greater than 800 were selected for analysis. The RI value was calculated according to the method in references [41,42]. Identification was performed by comparing the mass spectrum with that of the NIST library and by comparison of RI (retention index) with RI of published literatures and online library (https://webbook.nist.gov/chemistry/cas-ser.html (accessed on 20 October 2021)). According to the total ion flow chromatogram (Appendix A), the relative content of each component in the sample was determined using the peak area normalization method. Excel (Microsoft Excel 2010, version 14.0.6106.5005) was used to organize the data, Prism (GraphPad Prism 8, version v8.0.2.263) was used to construct the principal component accumulation diagram of the data, TBtools (Toolbox for Biologists, version v1.03) was used to perform the hierarchical cluster analysis of the data, and SPSS (IBM SPSS Statistics 22.0, version 9.5.0.0) was used for principal component analysis, the calculation of the eigenvector load value, and the study of the main floral fragrance components of *P. notoginseng*.

## 5. Conclusions

A total of 53 floral fragrance compounds were identified from different varieties of *P. notoginseng* in the forest, among which terpenes, alkynes, aromatic hydrocarbons, and alcohols were the main components of the floral aroma of *P. notoginseng* from the forest understory, while high contents of 3-carene, germacrene D, (−)-*α*-gurjunene, and valencene were identified in *P. notoginseng* flowers from the forest understory. (+)-*γ*-Gurjunene menogene and aromandendrene were also detected. The results showed that these substances were typical for a number of different *P. notoginseng* flowers in the forest. Cluster analysis showed that the main floral aroma components of S1 and S2 (four-forked seven leaf and three-forked seven leaf) were similar, while S4 (three-forked five–six leaf) were different in terms of main components. In addition, principal component analysis showed that the main substances distinguishing the four varieties of *P. notoginseng* were (+)-*γ*-gurjunene, (+)-calarene, copaene, 1,8,12-bisabolatriene, *γ*-elemene, (−)- aristolene, caryophyllene, 3-carenes, and 2,6-dimethyl-1,3,6-heptatriene.

## Figures and Tables

**Figure 1 molecules-27-03565-f001:**
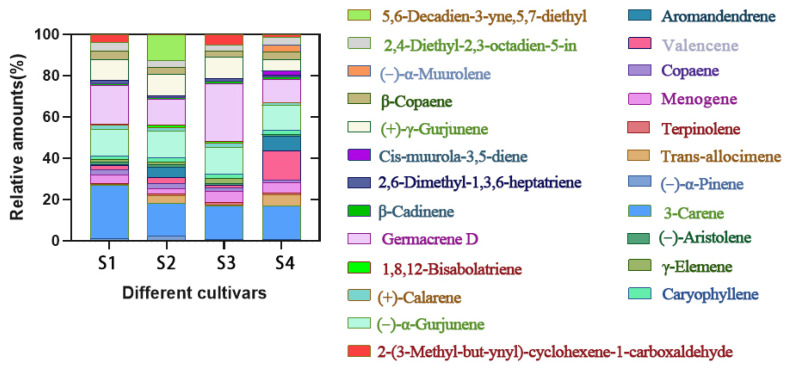
Floral fragrance compounds in the flowers of *P. notoginseng* (>1%).

**Figure 2 molecules-27-03565-f002:**
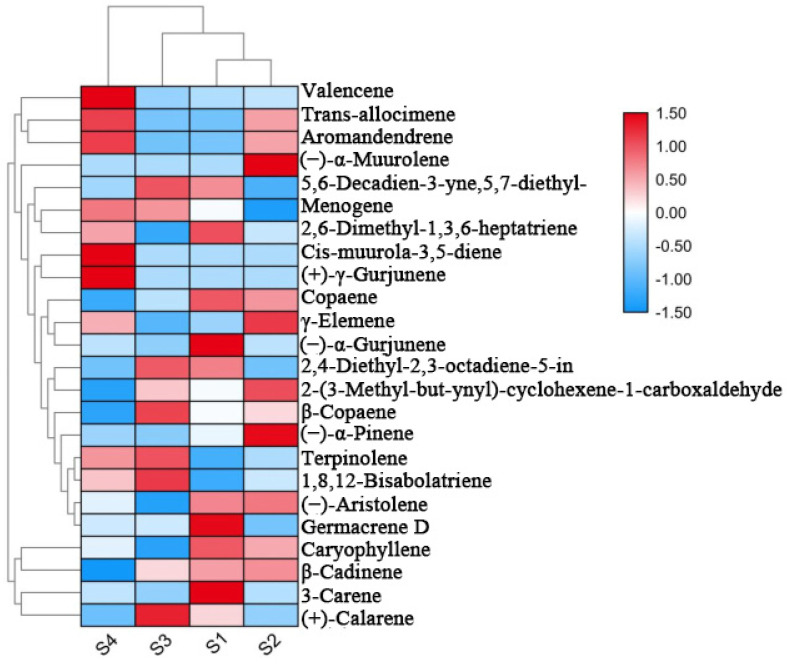
Heat map and hierarchical cluster analysis of floral fragrance compounds in the flowers of *P. notoginseng* (>1%).

**Figure 3 molecules-27-03565-f003:**
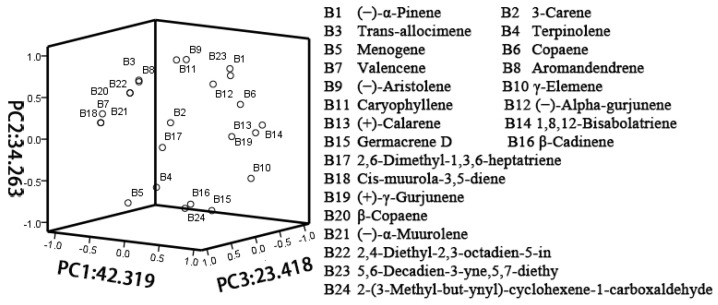
A 3D loading diagram of eigenvector loading values of 24 volatile components from PC1, PC2, and PC3, from different varieties of *P. notoginseng* flowers.

**Figure 4 molecules-27-03565-f004:**
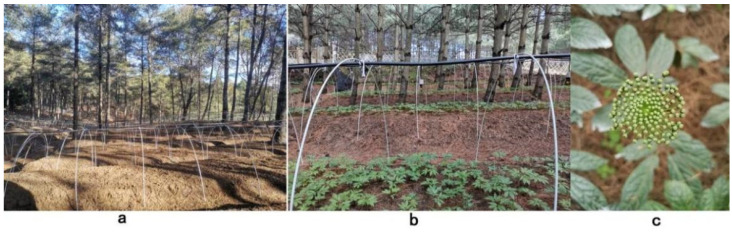
Schematic diagram of the *P. notoginseng*-*pinus* agroforestry system planting pattern. (**a**): woodland not planted with *Panax ginseng*; (**b**): woodland planted with *Panax ginseng*; and (**c**): *Panax ginseng* flower.

**Figure 5 molecules-27-03565-f005:**
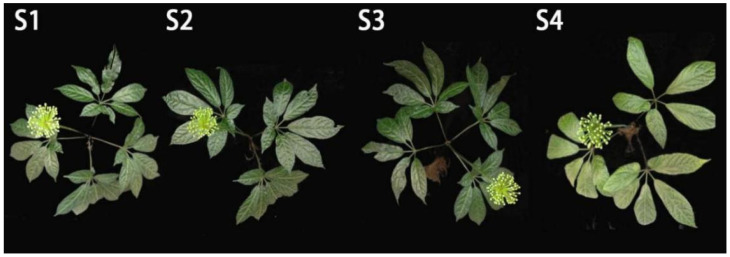
Morphological characteristics of flowers from different *P. notoginseng* varieties.

**Table 1 molecules-27-03565-t001:** Category and content of floral fragrance compounds in the flowers of different *P. notoginseng* varieties.

	Content and Type Number (%) ± SD and Type Number			
Category	S1	S2	S3	S4
Terpenes	80.05 ± 5.70b (25) ^1^	71.65 ± 0.47d (26)	74.75 ± 2.14c (26)	84.16 ± 7.13a (26)
Alcohols	0.31 ± 0.05b (1)	1.15 ± 0.26a (4)	0.50 ± 0.11b (2)	0.49 ± 0.30b (2)
Esters	- ^2^	-	0.34 ± 0.08 (2)	-
Aromatic hydrocarbons	10.61 ± 0.08a (3)	10.53 ± 0.10a (2)	9.50 ± 1.12ab (2)	8.87 ± 0.31b (3)
Alkynes	4.04 ± 0.08b (1)	15.85 ± 0.35a (2)	2.82 ± 0.59c (2)	3.73 ± 0.18b (1)
Alkanes	0.88 ± 0.33 (1)	-	1.00 ± 0.38 (1)	-
Aldehydes	3.97 ± 1.61a (1)	-	4.71 ± 0.86a (1)	1.20 ± 0.16b (1)
Others	0.11 ± 0.04c (3)	0.18 ± 0.05b (2)	1.17 ± 0.02a (3)	0.19 ± 0.01b (2)
Total ^3^	99.97 ± 8.32a (35)	99.36 ± 1.23a (36)	94.79 ± 5.34b (39)	98.54 ± 8.31a (35)

^1.^ The number of compound types. ^2.^ Not detected or nonexistent. ^3.^ The sum content of total compounds. Lowercase letters indicate significant differences at the *p* ≤ 0.05 level.

**Table 2 molecules-27-03565-t002:** Relative contents of floral fragrance compounds in the flowers of different *P. notoginseng* varieties.

Classification	Compound Name	RI ^1^	Relative Contents (%) ± SD			
			S1	S2	S3	S4
Terpenes	(−)-*α*-Pinene	937	1.15 ± 0.91a	2.16 ± 0.20a	0.76 ± 0.35a	0.83 ± 0.67a
	3-Carene	1011	25.68 ± 0.30a	15.87 ± 0.21b	14.82 ± 1.47b	16.23 ± 0.10b
	1,3-Cyclohexadiene,1,3,5,5-tetraMethyl-	1292	0.72 ± 0.59a	0.48 ± 0.13a	0.43 ± 0.19a	0.50 ± 0.41a
	2,6-Dimethyl-2,4,6-octatriene	1144	0.36 ± 0.11c	3.72 ± 0.15b	0.44 ± 0.17c	5.04 ± 0.58a
	Terpinolene	1088	0.64 ± 0.15c	0.84 ± 0.42bc	1.31 ± 0.02a	1.19 ± 0.15ab
	Menogene	1086	4.11 ± 0.40ab	2.54 ± 0.16b	4.88 ± 1.82a	4.72 ± 0.15a
	Copaene	1376	2.64 ± 0.08a	2.57 ± 0.04a	1.77 ± 0.02b	1.15 ± 0.07c
	Valencene	1492	2.08 ± 0.34b	2.73 ± 0.35b	1.11 ± 0.21c	14.23 ± 0.53a
	Aromandendrene	1440	0.19 ± 0.09c	5.10 ± 0.22b	0.08 ± 0.04c	7.02 ± 0.01a
	(−)-Aristolene	1453	1.20 ± 0.20a	1.22 ± 0.06a	0.69 ± 0.07b	0.98 ± 0.19a
	*β*-Maaliene	1405	0.30 ± 0.14a	0.32 ± 0.18a	0.24 ± 0.10a	0.39 ± 0.07a
	*γ*-Elemene	1433	1.24 ± 0.12c	1.50 ± 0.15b	2.34 ± 0.10a	- ^2^
	Caryophyllene	1419	1.91 ± 0.16a	2.04 ± 0.08a	1.88 ± 0.19a	1.99 ± 0.31a
	(−)-*α*-Gurjunene	1409	12.65 ± 0.30a	12.52 ± 0.15a	12.01 ± 1.67a	12.04 ± 0.69a
	(+)-Calarene	1432	2.04 ± 0.13a	1.77 ± 0.75a	1.73 ± 0.30a	1.37 ± 0.79a
	1,8,12-Bisabolatriene	-	0.69 ± 0.14b	1.27 ± 0.21a	0.89 ± 0.07b	-
	Isoledene	1375	0.16 ± 0.04a	0.19 ± 0.10a	0.27 ± 0.03a	-
	Germacrene D	1481	18.67 ± 0.30b	12.42 ± 0.07c	25.83 ± 0.17a	11.30 ± 0.91d
	*β*-Cadinene	1518	0.69 ± 0.44b	0.82 ± 0.08b	1.33 ± 0.08a	0.92 ± 0.20ab
	2-Pinene	937	0.62 ± 0.09a	0.25 ± 0.06b	0.42 ± 0.10ab	0.31 ± 0.23b
	*α*-Terpinene	1017	0.12 ± 0.06a	0.04 ± 0.01a	0.07 ± 0.05a	0.06 ± 0.05a
	(−)-Limonene	1031	0.26 ± 0.14a	0.17 ± 0.05a	0.22 ± 0.08a	0.16 ± 0.12a
	*γ*-Terpinene	1060	0.22 ± 0.20a	0.06 ± 0.02a	0.10 ± 0.04a	0.07 ± 0.06a
	2,6-Dimethyl-1,3,6-heptatriene	-	1.68 ± 0.32a	0.87 ± 0.28a	1.06 ± 0.84a	1.06 ± 0.77a
	Cosmene	1131	0.03 ± 0.02a	0.04 ± 0.03a	0.03 ± 0.02a	0.06 ± 0.03a
	Alpha-thujene	929	-	-	0.04 ± 0.03	-
	Alloaromadendrene	1461	-	0.14 ± 0.02	-	0.17 ± 0.06
	*α*-Guaiene	1439	-	-	-	0.26 ± 0.07
	(+)-Longifolene	1405	-	-	-	0.10 ± 0.06
	Cis-muurola-3,5-diene	1454	-	-	-	2.01 ± 0.03
Alcohols	(2R,4R,5S,6S,7R)-5-Isopropyl-2,8-dimethyltricyclo [4.4.0.02,7] decan-4-ol	-	0.31 ± 0.05a	0.38 ± 0.02a	0.41 ± 0.05a	0.27 ± 0.14a
	Spathulenol	1576	-	0.52 ± 0.16a	0.09 ± 0.06b	0.22 ± 0.16b
	Cubebol	1515	-	0.15 ± 0.02	-	-
	10-Epi-g-eudesmol	1619	-	0.10 ± 0.06	-	-
Esters	Vinyl myristate	1784	-	-	0.04 ± 0.03	-
	Phthalic acid,2-pentyl propyl ester	-	-	-	0.30 ± 0.05	-
Aromatic hydrocarbons	(+)-*γ*-Gurjunene	1473	10.28 ± 0.35a	10.49 ± 0.07a	9.47 ± 1.10a	5.25 ± 0.18b
	(+)-*δ*-Cadinene	1524	0.05 ± 0.03	-	-	-
	(+)-*γ*-Cadinene	1513	0.28 ± 0.09	-	-	-
	*α*-Calacorene	1542	-	0.04 ± 0.03	-	0.06 ± 0.04
	(−)-*α*-Muurolene	1485	-	-	-	3.56 ± 0.09
	Eudesma-3,7(11)-diene	1542			0.03 ± 0.02	
Alkynes	2,4-Diethyl-2,3-octadiene-5-in	-	4.04 ± 0.08a	3.28 ± 0.17bc	2.79 ± 0.58c	3.77 ± 0.18ab
	2-Methylnon-1-en-3-yne	-			0.03 ± 0.01	
	5,6-Decadien-3-yne,5,7-diethyl-	-	-	12.57 ± 0.18	-	-
Alkanes	*β*-Copaene	1432	0.88 ± 0.33	-	1.00 ± 0.38	-
Aldehydes	2-(3-Methyl-but-ynyl)-cyclohexene-1-carboxaldehyde	-	3.97 ± 1.61a	-	4.71 ± 0.86a	1.20 ± 0.16b
Others	4-Phenylsemicarbazide	-	0.03 ± 0.03	-	-	-
	3,5-Dimethylamphetamine	-	0.04 ± 0.02a	-	0.03 ± 0.02a	0.04 ± 0.02a
	2-Methoxy-3-Methylpy	954	0.04 ± 0.03	-	-	0.05 ± 0.03
	1-Hydroxymethyl-5,8,9-endo-tetramethyltricyclo	-	-	-	0.99 ± 0.01	-
	2,4,4-Trimelthyl-3-hydroxymethyl-5a-(3-methyl-but-2-enyl)-cyclohexene	-	-	0.14 ± 0.03	0.15 ± 0.03	-
	2-Isobutyl-3-methoxypyrazine	1183	-	0.04 ± 0.02	-	-
Total ^3^	53		35	36	39	35

^1.^ The values of the Kovats retention index. ^2.^ Not detected or nonexistent. ^3.^ Number of compounds. Lowercase letters indicate significant differences at the *p* ≤ 0.05 level.

## Data Availability

Not applicable.

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
