# Peer review of "Comparative Analysis of the Floral Fragrance Compounds of Panax notoginseng Flowers under the Panax notoginseng-pinus Agroforestry System Using SPME-GC-MS"

_molecules, 2022, doi:10.3390/molecules27113565_

Round 1

Reviewer 1 Report

Dear Authors,

You still did not lesson to my suggestions,

The text should be revised in terms of writing

I urge the authors to improve the English language for a better flow of literature. The manuscript needs revision regarding the language used.

The results need to revise in the text with tables.

Author Response

You still did not lesson to my suggestions,

The text should be revised in terms of writing

I urge the authors to improve the English language for a better flow of literature. The manuscript needs revision regarding the language used.

Thanks to the experts' suggestions and revisions, this article has been revised by MDPI's professional people for English, and I also checked the revisions of this article carefully. I hope the experts will be satisfied. Also, thank you for your contribution to the publication of this article.

Reviewer 2 Report

The authors undoubtedly did a great job of improving their manuscript, I can note the significantly increased quality of the presentation of the material.

But there are a small number of typos in the text.

Line 123: "components of in flowers" - please, delete "of"

Line 137: the same typos - need to delete the "of"

Line 161: different font sizes or different fonts are used for the fragment "β-cadinene, caryophyllene, (-)-aristolene, and copaene".

Line 163: please, use the small letters for written the compounds names in middle of sentence (Cis-muurola-3,5-diene, (+)-γ-Gurjunene and (+)-Calarene). 

Line 258: please, use the small letters for written the compounds names in middle of sentence (Spatulenol).

Thank for your efforts and good luck! 

Author Response

The authors undoubtedly did a great job of improving their manuscript, I can note the significantly increased quality of the presentation of the material.

Thanks to the experts' suggestions and revisions, this article has been revised by MDPI's professional people for English, and I also checked the revisions of this article carefully. I hope the experts will be satisfied. Also, thank you for your contribution to the publication of this article.

But there are a small number of typos in the text.

Line 123: "components of in flowers" - please, delete "of"

We have modified it.

Line 137: the same typos - need to delete the "of"

We have modified it.

Line 161: different font sizes or different fonts are used for the fragment "β-cadinene, caryophyllene, (-)-aristolene, and copaene".

We have uniformed it.

Line 163: please, use the small letters for written the compounds names in middle of sentence (Cis-muurola-3,5-diene, (+)-γ-Gurjunene and (+)-Calarene). 

We have modified it.

Line 258: please, use the small letters for written the compounds names in middle of sentence (Spatulenol).

We have modified it.

Thank for your efforts and good luck! 

Round 2

Reviewer 1 Report

I thank the authors for their efforts, there are two important points:

Point1: Please describe the IR method in more detail (section data analysis). please check this paper, https://doi.org/10.1016/j.jchromb.2013.11.032

Point 2: The results in the text are not the same in the tables; they need to be revised. For example, they wrote in the text terpenes (72.11–85.41%) but in table 1 (71.65–84.16%), all the data should check and revise.

Author Response

Thank you to the experts for their helps in making this article more readable.

1 Please describe the IR method in more detail (section data analysis). please check this paper, https://doi.org/10.1016/j.jchromb.2013.11.032

Reply: Thank you very much for your suggestions. We have carefully reviewed the relevant literature and added relevant descriptions in the section data analysis.

2 The results in the text are not the same in the tables; they need to be revised. For example, they wrote in the text terpenes (72.11–85.41%) but in table 1 (71.65–84.16%), all the data should check and revise.

Reply: Sorry, this is a mistake in our work. We have carefully checked the data of table1 and corrected the error. At the same time, we have checked other data to ensure its accuracy.

This manuscript is a resubmission of an earlier submission. The following is a list of the peer review reports and author responses from that submission.

Round 1

Reviewer 1 Report

This paper devoted to comparison of volatile components composition of four kinds Panax notoginseng flowers by headspace solid phase microextraction and GC/MS methods. 

The objectives and results of the study raise a number of questions. 

First of all, the topic and object of study seem very hackneyed. The applied significance of the results is not clear. The conclusions seem like a statement of facts, moreover, some of the wording is failed. The text of this manuscript contains some mistakes and blots.

In general, the article looks weak and incomplete. Unfortunately, I have to reject this manuscript. 

Please, find the more detailed notes in attached file. 

Author Response

Reviewer1

This paper devoted to comparison of volatile components composition of four kinds Panax notoginseng flowers by headspace solid phase microextraction and GC/MS methods. 

The objectives and results of the study raise a number of questions. 

First of all, the topic and object of study seem very hackneyed. The applied significance of the results is not clear. The conclusions seem like a statement of facts, moreover, some of the wording is failed. The text of this manuscript contains some mistakes and blots.

In general, the article looks weak and incomplete. Unfortunately, I have to reject this manuscript. 

Please, find the more detailed notes in attached file. 

Reply:

1, Thank you very much for your comment, but what I want to tell you is that wild Panax notoginseng originated in the forest, but now wild Panax notoginseng no longer exists. This experiment takes Panax notoginseng in agro-forestry system as the research object to analyze the aroma components of Panax notoginseng flower, hence the material is novel.

The traditional field planting method adopts large amount of water, fertilizer and pesticides, while Panax notoginseng in agro-forestry system adopts the method of no fertilization and imitates the planting mode of wild Panax notoginseng. At present, the previous researches only focused on the volatile oil of Panax Notoginseng flower, and no one has studied the aroma components the flower of Panax Notoginseng. Therefore, we think it is very important to detect the floral aroma components of Panax notoginseng in agro-forestry system.

2, Other details have been improved according your suggestions in the manuscript.

Reviewer 2 Report

No título coloque o nome da família após a espécie.

REMOVER DAS PALAVRAS-CHAVE:  Panax notoginseng da floresta. Coloque Panax ginseng

RESUMO: Colocar gamma gurjunene em minúsculas

INTRODUÇÃO

Cite os outros trabalhos realizados com as flores de Panax notoginseng, descrevendo os compostos químicos isolados e as atividades farmacológicas encontradas.

Existem artigos com produção e identificação de compostos voláteis com a espécie? E quanto ao gênero ou família? Cite-os e compare os resultados encontrados.

Tabela 1 - Com os dados da espectrometria de massas, não foi possível identificar todos os compostos?

Faça uma tabela com os dados e fragmentos de espectrometria de massa

L.109 - -γ-GURHUNENE – corrigir a palavra

L.139 - colocar variedades após quatro

L.148 – A Figura 2 não foi citada no texto

L.173 – ele resulta??

L.238 – duas formas de flor foram numeradas S1, S2, S3 e S4. Explicar.

L.289 - ...valência e ??

Na discussão mostra-se que já foram realizadas análises de voláteis com as mesmas espécies coletadas em outras regiões. Então, por que esse tipo de análise foi feito novamente?

Você tem testes de atividade farmacológica para comparar os efeitos? Outros trabalhos possuem análises quimiométricas que possam mostrar o perfil dos compostos identificados? Faz pouco sentido repetir essas análises sem mostrar a importância ou um próximo passo para a pesquisa. Deixe isso mais claro no texto.

Author Response

Comments and Suggestions for Authors

Você tem testes de atividade farmacológica para comparar os efeitos? Outros trabalhos possuem análises quimiométricas que possam mostrar o perfil dos compostos identificados? Faz pouco sentido repetir essas análises sem mostrar a importância ou um próximo passo para a pesquisa. Deixe isso mais claro no texto.

1 No título coloque o nome da família após a espécie.

Reply: The article name has been modified: Comparative Analysis of the Floral Fragrance Compounds of Panax notoginseng Flowers under the Panax notoginseng-pinus Agroforestry System by SPME-GC/MS

2 REMOVER DAS PALAVRAS-CHAVE:  Panax notoginseng da floresta. Coloque Panax ginseng

Reply: Keywords have been deleted.

3 RESUMO: Colocar gamma gurjunene em minúsculas

Reply: it has been modified.

4 Cite os outros trabalhos realizados com as flores de Panax notoginseng, descrevendo os compostos químicos isolados e as atividades farmacológicas encontradas.

Other studies of Panax notoginseng flower and their pharmacological effects have been added in the introduction.

5 Existem artigos com produção e identificação de compostos voláteis com a espécie? E quanto ao gênero ou família? Cite-os e compare os resultados encontrados.

Reply: The aroma components of Panax notoginseng flower detected in our study, previous studies focused on the volatile oil of Panax Notoginseng flower.

6 Tabela 1 - Com os dados da espectrometria de massas, não foi possível identificar todos os compostos?

Reply: Floral Fragrance Compounds analyzed with NIST14 mass spectrometry database according to headspace solid phase microextraction method.

7 Faça uma tabela com os dados e fragmentos de espectrometria de massa

Reply: Mass spectrometry total ion flow diagrams have been provided.

8 L.109 - -γ-GURHUNENE – corrigir a palavra

Reply: Has been modified

9 L.139 - colocar variedades após quatro

Reply: Has been modified

10 L.148 – A Figura 2 não foi citada no texto

Reply: Has been modified

11 L.173 – ele resulta??

Reply: Has been modified

12 L.238 – duas formas de flor foram numeradas S1, S2, S3 e S4. Explicar.

Reply: Has been modified

13 L.289 - ...valência e ??

Reply: Has been modified

14 Na discussão mostra-se que já foram realizadas análises de voláteis com as mesmas espécies coletadas em outras regiões. Então, por que esse tipo de análise foi feito novamente?

Reply: The volatile oil components of Panax notoginseng flower are mentioned in the discussion, and their identification years are earlier, and the methods and equipment also have limitations. Therefore, we use SPME-GC-MS to identify the aroma components of Panax notoginseng flower. In additon, we discussed on the difference between floral fragance compounds and oil volatile compounds in the paper.

15 Você tem testes de atividade farmacológica para comparar os efeitos? Outros trabalhos possuem análises quimiométricas que possam mostrar o perfil dos compostos identificados? Faz pouco sentido repetir essas análises sem mostrar a importância ou um próximo passo para a pesquisa. Deixe isso mais claro no texto.

Reply: We have added the sentence: In the next step, we will analyze and test the pharmacological activities of the main floral fragrance compounds of Panax notoginseng flower, so as to provide a basis for future application.

Reviewer 3 Report

The article is interesting, but it does contain some inaccuracies. They concern:
title: should be a bit shorter, in its current form it is quite illegible.
This is also related to the suggestion for keywords, which unfortunately are a repetition of the title. In addition, they should be phrases, the most important keys that the reader can use to find the article in search engines. However, they should not duplicate the title. The authors are asked to correct these inaccuracies.
Abstract:
It is unnecessary to inform in this part of the article that flowers have a healing value, since the authors do not indicate what this property is and in what ailments they are used or recommended. We do not know whether it is flowers or extracts or preparations with this material.
Introduction: It is quite laconic and contains basic information, generally rather popular science. There are no significant issues that would make the reader meaningful to the experiment performed with respect to the relevant research on the scope of the topic. It is unclear whether research into variety diversity has been conducted before. This information must be included. Likewise, it would be worth mentioning the size of the cultivation of the varieties in question. We do not know anything about it, therefore it should be completed.
The authors also did not indicate the novelty of their experiment. How does the presented research differ from the previous ones that have already been published? So, please outline the background around the analyzed raw material.

Author Response

1 title: should be a bit shorter, in its current form it is quite illegible.

Reply: The article title has been modified

2 This is also related to the suggestion for keywords, which unfortunately are a repetition of the title. In addition, they should be phrases, the most important keys that the reader can use to find the article in search engines. However, they should not duplicate the title. The authors are asked to correct these inaccuracies.
Reply:The keywords in the article have been modified

3 It is unnecessary to inform in this part of the article that flowers have a healing value, since the authors do not indicate what this property is and in what ailments they are used or recommended. We do not know whether it is flowers or extracts or preparations with this material.
Reply: We have modified the sentences: The flower of P. notoginseng has a valuable medicinal material and its functions in clearing away heat and detoxifying, calms the liver and improves eyesight, and produces saliva and quenching thirst . These functions are mainly due to the variety of active components contained in Panax notoginseng flowers, such as: olefin compounds, ginsenoside (Rb1, RC, RB3, Rb2 and Rd) and polyphenol.

4 Introduction: It is quite laconic and contains basic information, generally rather popular science. There are no significant issues that would make the reader meaningful to the experiment performed with respect to the relevant research on the scope of the topic. It is unclear whether research into variety diversity has been conducted before. This information must be included. Likewise, it would be worth mentioning the size of the cultivation of the varieties in question. We do not know anything about it, therefore it should be completed.

Reply: The planting scale has been supplemented in the introduction. After 2015, the agroforestry model of Pine-Sanqi has been planted in large quantities in Yunnan (Pu'er, Lincang, Kunming, Qujing, etc.) with an area of about 1500 ha.

5 The authors also did not indicate the novelty of their experiment. How does the presented research differ from the previous ones that have already been published? So, please outline the background around the analyzed raw material.

Reply: Thank you for your objective evaluation. At present, there has been no study on variety diversity of Panax notoginseng. In addition, we have added an introduction to the planting scale of varieties in the preface.

Reviewer 4 Report

The reviewed paper concerns the volatile components of Panax notoginseng flowers of four varieties were compared and analyzed by headspace solid phase microextraction GC/MS. Largely the work performed is of good quality and conclusions are supported by the data. 

Below, there is a list of suggestions that in my opinion would help to improve the manuscript.

  1. Page 11 line 204 instead of (+)-γ-GURHUNENE please write (+)-γ-gurjunene, and through the whole the text please change the capital letters to the small ones
  2. As well, γ should be italic through the whole text and in tables

Author Response

The reviewed paper concerns the volatile components of Panax notoginseng flowers of four varieties were compared and analyzed by headspace solid phase microextraction GC/MS. Largely the work performed is of good quality and conclusions are supported by the data. 

Below, there is a list of suggestions that in my opinion would help to improve the manuscript.

Thank you for your comments on this manuscript. We have revised it according to the suggestions of other reviewers. Please check it.

1, Page 11 line 204 instead of (+)-γ-GURHUNENE please write (+)-γ-gurjunene, and through the whole the text please change the capital letters to the small ones

Reply: It has been corrected to the correct format in the full text.

2, As well, γ should be italic through the whole text and in tables

Reply: The full text has been γ correct all to italics.

Reviewer 5 Report

Dear Authors,

The studies are interesting, but the manuscript needs more improvement before eventual publication. Below are some queries.

The text should be revised in terms of writing

I urge the authors to improve the English language for a better flow of literature. The manuscript needs revision regarding the language used.

The results need to revise in the text with tables

Title:

Revise the title and add “SPME-GC-MS”; under forest using SPME-GC-MS

Keywords:

Remove from the forest

Results

Line 94, Authors should add a sentence to describe the meaning of s1, s2, s3, s4

Line 94, eight volatiles what? Please add “groups” eight volatiles groups

Line 98, respectively???

Table 1

Change to compounds category

Content? is it relative content?

Please add the significant letters (a, b, c, d) between the flower types

What does the superscript mean (1) and (2)? Please mention in the table footnote

Type number? Sometimes you write the first letter capital; please use one style

The results in the text are not the same in the tables; they need to be revised.

The authors should add the significant letters (a, b, c, d) between the flower types (table 2)

Total? What does it mean? Please add in the footnote

The qualitative analysis of volatile compounds is not accurate only by MS, so it is suggested to increase RI for further qualitative analysis.

The authors should add the odour description of the volatile compounds

The sections of results and discussion are very weak. Authors should pay more attention to the results and discussion section in scientific discussion and revise the results with tables “many data not accurate.”

Line 249, 20 ml? we usually use 25 ml

Please revise the HS-SPME analysis and GC-MS analysis method and add the references

The split ratio?

I need to see the total ion current chromatograms of volatile compounds detected by the SPME-GC‐ MS technique, and also you can put them as the supplementary data file

Author Response

Thank you for your suggestions, we have modified the paper.

1, Revise the title and add “SPME-GC-MS”; under forest using SPME-GC-MS

Reply: The article title has been modified.

2, Remove from the forest

Reply: Keywords have been modified.

3, Line 94, Authors should add a sentence to describe the meaning of s1, s2, s3, s4

Reply: Description has been added for S1, S2, S3, S4.

4, Line 94, eight volatiles what? Please add “groups” eight volatiles groups

Reply: Thanks, the description here is incorrect and has been modified.

5, Change to compounds category. Content? is it relative content?

Reply: It has been changed to compound category, the contents in the table are corresponding, and corresponding a, b, c and d have been added.

6, What does the superscript mean (1) and (2)? Please mention in the table footnote

Reply: Footnotes have been added.

7, Type number? Sometimes you write the first letter capital; please use one style

Reply: All initials have been modified.

8, The results in the text are not the same in the tables; they need to be revised.

Reply: Inaccuracies in the results have been corrected.

9, The authors should add the significant letters (a, b, c, d) between the flower types (table 2)

Reply: Corresponding a, b, c and d have been added.

10, Total? What does it mean? Please add in the footnote

Reply:Footnotes have been added.

11,The qualitative analysis of volatile compounds is not accurate only by MS, so it is suggested to increase RI for further qualitative analysis.

Reply: RI has been added.

12, The authors should add the odour description of the volatile compounds.

We have added the sentence: The flower of P. notoginseng in conventionally agricultrual system has a slightly bitter odour, but the flowers of P. notoginseng in the agroforestry system have different odour, which maybe due to the enviroment factors result in producing some special compounds.

13, The sections of results and discussion are very weak. Authors should pay more attention to the results and discussion section in scientific discussion and revise the results with tables “many data not accurate.”

Reply:The accuracy of the data has been improved. And the results and discussions have been tried to improve.

14, Line 249, 20 ml? we usually use 25 ml

Reply: We have confirmed that the volume of the bottle is 20ml.

15, Please revise the HS-SPME analysis and GC-MS analysis method and add the references,The split ratio?

Reply: Materials and methods have been modified and references have been added.

16, I need to see the total ion current chromatograms of volatile compounds detected by the SPME-GC-MS technique, and also you can put them as the supplementary data file

Reply: We have added the total ion flow diagram in the appendix.

Round 2

Reviewer 1 Report

Dear authors,

Thank you for your response, but form of this one is not so correct. 

Your response have to consist of step-by-step answers of reviewer's questions. And you had not adhered to this form. This is a very big omission on your part.

Next, some of my corrections were still ignored by you.

I hope, that your work after more thorough editing can be accepted into the "Metabolites" journal. 

Sorry, but I have to put "reject". 

Reviewer 5 Report

Dear Authors,

You still did not lesson to my suggestions,

The text should be revised in terms of writing

I urge the authors to improve the English language for a better flow of literature. The manuscript needs revision regarding the language used.

The results need to revise in the text with tables

Results

Line 94, Authors should add a sentence to describe the meaning of s1, s2, s3, s4

Line 94, eight volatiles what? Please add “groups” eight volatiles groups

Line 98, respectively???

Table 1

Content? is it relative content?

Please add the significant letters (a, b, c, d) between the flower types, you did but the statistical analysis is wrong

Type number? Sometimes you write the first letter capital; please use one style

The results in the text are not the same in the tables; they need to be revised.

The authors should add the significant letters (a, b, c, d) between the flower types (table 2) you did but the statistical analysis is wrong

Total of what? What does it mean? Please add in the footnote (sum of total compounds content)

The qualitative analysis of volatile compounds is not accurate only by MS, so it is suggested to increase RI for further qualitative analysis. I know you added RI number, until now I afraid you do know what is RI? And how you calculate it? Where is the chromatograph of the n-series alkanes? Show me, moreover the total ion current chromatograms of volatile compounds detected by the SPME-GC‐ MS technique, I see they are not real. (All the diagram same!!!!!!!?????

The authors should add the odour description of the volatile compounds. You should describe the odour on the table

The sections of results and discussion are very weak. Authors should pay more attention to the results and discussion section in scientific discussion and revise the results with tables “many data not accurate.”

Please revise the HS-SPME analysis and GC-MS analysis method and add the references, you did not revise, also where is the method for the identified the volatile compounds based on RI?

The split ratio?

I need to see the total ion current chromatograms of volatile compounds detected by the SPME-GC‐ MS technique, and also you can put them as the supplementary data file. They are not real

The references are not in the journal format. (Abbreviated Journal Name, where is it?)
